

# Mahalanobis distances for ecological niche modelling and outlier detection: implications of sample size, error, and bias for selecting and parameterising a multivariate location and scatter method

Thomas R. Etherington

Manaaki Whenua –Landcare Research, Lincoln, New Zealand

Corresponding author
Thomas R. Etherington, etherington@landcareresearch.co.nz

## ABSTRACT

The Mahalanobis distance is a statistical technique that has been used in statistics and data science for data classification and outlier detection, and in ecology to quantify species-environment relationships in habitat and ecological niche models. Mahalanobis distances are based on the location and scatter of a multivariate normal distribution, and can measure how distant any point in space is from the centre of this kind of distribution. Three different methods for calculating the multivariate location and scatter are commonly used: the sample mean and variance-covariance, the minimum covariance determinant, and the minimum volume ellipsoid. The minimum covariance determinant and minimum volume ellipsoid were developed to be robust to outliers by minimising the multivariate location and scatter for a subset of the full sample, with the proportion of the full sample forming the subset being controlled by a user-defined parameter. This outlier robustness means the minimum covariance determinant and the minimum volume ellipsoid are highly relevant for ecological niche analyses, which are usually based on natural history observations that are likely to contain errors. However, natural history observations will also contain extreme bias, to which the minimum covariance determinant and the minimum volume ellipsoid will also be sensitive. To provide guidance for selecting and parameterising a multivariate location and scatter method, a series of virtual ecological niche modelling experiments were conducted to demonstrate the performance of each multivariate location and scatter method under different levels of sample size, errors, and bias. The results show that there is no optimal modelling approach, and that choices need to be made based on the individual data and question. The sample mean and variance-covariance method will perform best on very small sample sizes if the data are free of error and bias. At larger sample sizes the minimum covariance determinant and minimum volume ellipsoid methods perform as well or better, but only if they are appropriately parameterised. Modellers who are more concerned about the prevalence of errors should retain a smaller proportion of the full data set, while modellers more concerned about the prevalence of bias should retain a larger proportion of the full data set. I conclude that Mahalanobis distances are a useful niche modelling technique, but only for questions relating to the fundamental niche of a species where the assumption of multivariate normality is reasonable. Users of the minimum covariance determinant and minimum

volume ellipsoid methods must also clearly report their parameterisations so that the results can be interpreted correctly.

## INTRODUCTION

The Mahalanobis distance (*Mahalanobis, 1936*) is a statistical technique that can be used to measure how distant a point is from the centre of a multivariate normal distribution. Mahalanobis distances are commonly applied to problems such as classifying data into groups and determining differences between groups (*Manly, 2005*). Mahalanobis distances have also been used to quantify species-environment relationships through habitat and ecological niche models (*Dettmers, Buehler & Bartlett, 2002*; *Johnson & Gillingham, 2005*; *Tsoar et al., 2007*; *Etherington et al., 2009*). In this context Mahalanobis distances are classified as a presence-only technique because they do not require species absence or background environmental data (*Peterson et al., 2011*) and simply require a data matrix

$$
\mathbf{A} = \begin{bmatrix}
x_{1,1} & x_{1,2} & x_{1,3} & \cdots & x_{1,n} \\
x_{2,1} & x_{2,2} & x_{2,3} & \cdots & x_{2,n} \\
x_{3,1} & x_{3,2} & x_{3,3} & \cdots & x_{3,n} \\
\vdots & \vdots & \vdots & \ddots & \vdots \\
x_{m,1} & x_{m,2} & x_{m,3} & \cdots & x_{m,n}
\end{bmatrix}
$$

that has $m$ rows of species occurrences for which $n$ columns of environmental variables have been obtained. The multivariate location and scatter of the data are defined by an $n$-dimensional vector $\hat{\boldsymbol{\mu}}$ containing the sample means for each column of variables, and a sample variance-covariance matrix $\hat{\boldsymbol{\Sigma}}$ of dimensions $n \times n$ that contains variances for each column along the main diagonal and pair-wise column covariances values elsewhere (*Manly, 2005*).

The Mahalanobis distance

$$
D^2(\mathbf{x}) = (\mathbf{x} - \hat{\boldsymbol{\mu}})^{\mathrm{T}} \hat{\boldsymbol{\Sigma}}^{-1} (\mathbf{x} - \hat{\boldsymbol{\mu}}) \tag{1}
$$

can then be calculated for any vector $\mathbf{x} = [x_1, x_2, x_3, \ldots, x_n]$ that represents a position in environmental space as defined by the $n$ environmental variables. As $D^2$ is essentially the sum of $n$ independent standard normal variables, the $D^2$ values from a multivariate normal population will follow a chi-squared distribution with degrees of freedom equal to the number of dimensions $n$ (*Manly, 2005*). This means that a chi-squared cumulative distribution function $F_{\chi_n^2}(x)$ can be used to convert $D^2$ into a probability $P(\chi_n^2 \leq D^2)$ that indicates if a location in environmental space has a $D^2$ that is greater than would be expected by chance (*Etherington, 2019*). For example, when applied to $m = 20$ hypothetical points in $n = 2$ dimensions, the calculated $P(\chi_n^2 \leq D^2)$ values follow a characteristic elliptical pattern
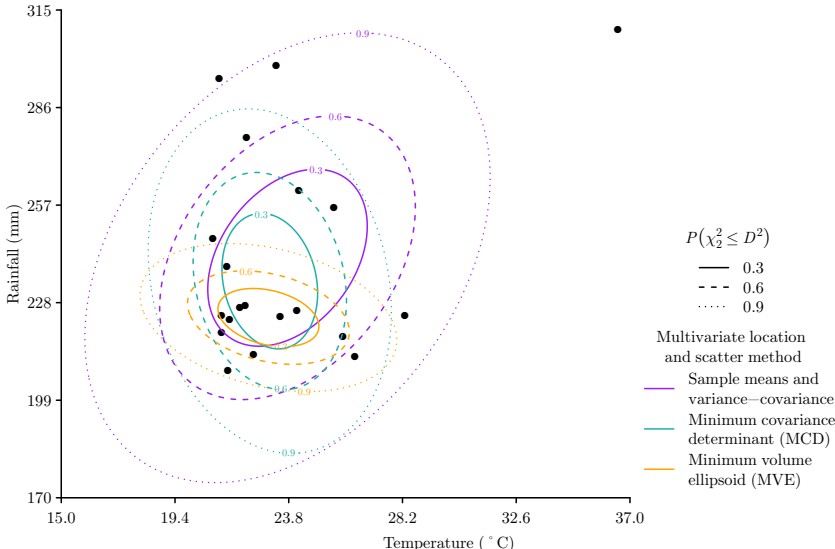

**Figure 1** **Hypothetical two-dimensional example of Mahalanobis distance $D^2$ with three different methods of defining the multivariate location and scatter of the data.** For each method the ellipses show contours of probability $P(\chi_n^2 \leq D^2)$ that indicate if a location in environmental space has a $D^2$ that is greater than would be expected by chance. The sensitivity of the sample mean and variance-covariance matrix method to outlying data can be seen, which contrasts with both the minimum covariance determinant and minimum volume ellipsoid methods, which both focus on where data is concentrated.

centred at the mean of each environmental variable, with $P(\chi_n^2 \leq D^2)$ increasing from the centre outwards in a manner that accounts for the variability within and correlation between each environmental variable (Fig. 1).

The elliptical form of Mahalanobis distances fits well with the theoretical concept of the fundamental niche, which *Hutchinson (1957)* p. 416 defined as "an *n*-dimensional hypervolume ... which corresponds to a state of the environment which would permit the species ... to exist indefinitely". *Hutchinson (1957)* used a rectangular model to define environment limits of the fundamental niche, but also stated that "If the variables are independent in their action on the species we may regard this area as the rectangle ... but failing such independence the area will exist whatever the shape of its sides". So any convex shape, which includes the elliptical shape of $P(\chi_n^2 \leq D^2)$, would be an appropriate model of the fundamental niche. Indeed, we see the use of ellipses alongside other convex shapes in later development of the niche concept (*Hutchinson, 1978*). In this context $\hat{\boldsymbol{\mu}}$ represents the optimal environmental conditions at the centre of the fundamental niche, and $\hat{\boldsymbol{\Sigma}}$ represents both the range of and interaction between environmental conditions within the fundamental niche.

While $P(\chi_n^2 \leq D^2)$ is usually inverted to $P(\chi_n^2 > D^2)$ for use in ecological niche modelling to estimate the probability of an environmental location being within a fundamental niche (*Etherington, 2019*), $P(\chi_n^2 \leq D^2)$ is also commonly used in statistics and data science to detect outliers (*Aggarwal, 2017*). However, when applied to detect outliers, the use of sample means and variance-covariance to estimate $D^2$ can be problematic because these

measures of multivariate location and scatter are sensitive to outliers (Fig. 1). So for outlier detection, $D^2$ can be calculated with different methods for defining the multivariate location and scatter of data, such as the minimum covariance determinant (MCD) and minimum volume ellipsoid (MVE), which are much more insensitive to outliers (*Rousseeuw, 1985*).

The MCD approach estimates multivariate location $\hat{\boldsymbol{\mu}}_{\mathrm{MCD}}$ and scatter $\hat{\boldsymbol{\Sigma}}_{\mathrm{MCD}}$ from a subset numbering $h$ data points that has the smallest variance-covariance matrix determinant (*Hubert & Debruyne, 2010*). The MVE approach is similar to the MCD in that it works with a subset of size $h$ data points, but the MVE estimates multivariate location $\hat{\boldsymbol{\mu}}_{\mathrm{MVE}}$ and scatter $\hat{\boldsymbol{\Sigma}}_{\mathrm{MVE}}$ from the ellipsoid of minimal volume that encapsulates the $h$ data points (*Van Aelst & Rousseeuw, 2009*). $D^2$ can then be calculated using either the MCD measures of multivariate location and scatter

$$D^2(\mathbf{x}) = (\mathbf{x} - \hat{\boldsymbol{\mu}}_{\mathrm{MCD}})^{\mathrm{T}} \hat{\boldsymbol{\Sigma}}_{\mathrm{MCD}}^{-1} (\mathbf{x} - \hat{\boldsymbol{\mu}}_{\mathrm{MCD}}) \tag{2}$$

or the MVE measures of multivariate location and scatter

$$D^2(\mathbf{x}) = (\mathbf{x} - \hat{\boldsymbol{\mu}}_{\mathrm{MVE}})^{\mathrm{T}} \hat{\boldsymbol{\Sigma}}_{\mathrm{MVE}}^{-1} (\mathbf{x} - \hat{\boldsymbol{\mu}}_{\mathrm{MVE}}) \tag{3}$$

for any vector $\mathbf{x} = [x_1, x_2, x_3, \ldots, x_n]$ that represents a position in $n$-dimensional environmental space.

The utility of the MCD and MVE methods for defining the multivariate location and scatter of data is highly relevant for ecological niche modelling based on digitally mobilised data through data-sharing networks such as the Global Biodiversity Information Facility (*Edwards, Lane & Nielsen, 2000*). These networks are reliant on natural history observation data that are likely to contain errors such as taxonomic misidentification or incorrect and imprecise georeferencing (*Graham et al., 2004*), which can result in species occurrences that are outliers in environmental space. Returning to our hypothetical example (Fig. 1), we can see that both the MCD and MVE methods ignore the apparent outlier and produce measures of multivariate location and scatter that are focussed on where the data are more concentrated. Given the robustness of MCD and MVE methods to outliers resulting from errors in natural history observation data, it is perhaps no surprise that both methods have been adopted recently for ecological niche modelling (*Norris, Jackson & Betancourt, 2006; Liu, White & Newell, 2018; Soberón, Peterson & Osorio-Olvera, 2018; Yañez-Arenas et al., 2018; Qiao et al., 2019; Altamiranda-Saavedra et al., 2020; Osorio-Olvera et al., 2020; Castaño-Quintero et al., 2020*).

On the other hand, natural history observation data will also contain sampling bias (*Graham et al., 2004*) that can be extreme and result in greater amounts of data for more charismatic species, more accessible places, more developed countries, and more recent times (*Meyer, Weigelt & Kreft, 2016*). If unaccounted for this could skew models results. Therefore, returning to our hypothetical example (Fig. 1), we may also have a situation in which the apparent outlier only appears to be an outlier due to sampling bias at environments with lower temperatures and rainfall. If this were the situation then the MCD and MVE methods would be providing a poorer estimate of multivariate location and scatter because they are both focussing on the data bias, whereas the method based
on the sample mean and covariance provides a better estimate that relates to all the data points.

Unfortunately, the differing effects of errors and bias means there is unlikely to be a best method for estimating the multivariate location and scatter of data in all situations, but this is consistent with ecological niche modelling more generally (*Qiao, Soberón & Peterson, 2015*). Therefore, this paper uses a virtual ecology approach (*Zurell et al., 2010*) to simulate a series of ecological niche modelling experiments to understand when different methods of defining the multivariate location and scatter of the data are more appropriate as sample size, errors, and bias vary.

## MATERIALS & METHODS

The virtual species used in the experiments is the Antipodean opaleye dragon, which is imagined to live in the mountain valleys of New Zealand (*Scamander, 2001*). I have defined the species' fundamental niche in terms of population growth rates (*Maguire, 1973*) measured as the finite rate of increase $\lambda_F$ of a population in a two-dimensional environmental space of temperature and rainfall. The fundamental niche is defined by three parameters: $\lambda_{\max}$, the maximum finite rate of increase at the fundamental niche optimum; $\boldsymbol{\mu}$, a $2 \times 1$ column vector of means that gives the optimal temperature and rainfall condition; and $\boldsymbol{\Sigma}$, a $2 \times 2$ variance-covariance matrix that determines the size and orientation of the fundamental niche in each dimension. With

$$\lambda_{\max} = 2.5, \boldsymbol{\mu} = \begin{bmatrix} 7.5 \\ 1800 \end{bmatrix}, \text{ and } \boldsymbol{\Sigma} = \begin{bmatrix} 2 & -950 \\ -950 & 800000 \end{bmatrix}$$

the fundamental niche finite rate of increase $\lambda_F(\mathbf{x})$ of a virtual species for any vector $\mathbf{x}$ of environmental space coordinates is then calculated as

$$\lambda_F(\mathbf{x}) = \lambda_{\max} \times e^{-\frac{1}{2}(\mathbf{x}-\boldsymbol{\mu})^{\mathrm{T}} \boldsymbol{\Sigma}^{-1}(\mathbf{x}-\boldsymbol{\mu})} \tag{4}$$

that results in an elliptically shaped fundamental niche (Fig. 2A).

Generating samples of species occurrences begins with an idealised virtual sampling of the niche space. An initial set of sampling locations $S = \{\mathbf{x}_1, \mathbf{x}_2, \ldots, \mathbf{x}_i\}$ consisted of a series of locations in environmental space. $S$ was randomly generated from a two-dimensional normal distribution $S \sim \mathcal{N}_2(\boldsymbol{\mu}, \boldsymbol{\Sigma})$ with the same vector of means $\boldsymbol{\mu}$ and variance-covariance matrix $\boldsymbol{\Sigma}$ as defined the fundamental niche. Then the probability that a sampled location $\mathbf{x}_i$ resulted in the species being both present and detected $P_d(\mathbf{x}_i)$ was described as a logistic function

$$P_d(\mathbf{x}_i) = \frac{1}{1 + e^{-10(\lambda_F(\mathbf{x}_i) - 0.5)}} \tag{5}$$

which was parameterised so that $P_d(\mathbf{x}_i)$ increases as $\lambda_F(\mathbf{x}_i)$ increases, but with $P_d(\mathbf{x}_i) \approx 0$ where $\lambda_F(\mathbf{x}_i) \approx 0$ because the virtual species population is unlikely to exist under these environmental conditions, and $P_d(\mathbf{x}_i) \approx 1$ where $\lambda_F(\mathbf{x}_i) \gtrsim 1$ because above this population growth rate the population should always be present.

Using this idealised sampling, a sample of $m = 100$ occurrence locations can be produced that are concentrated towards the centre of the fundamental niche, and for
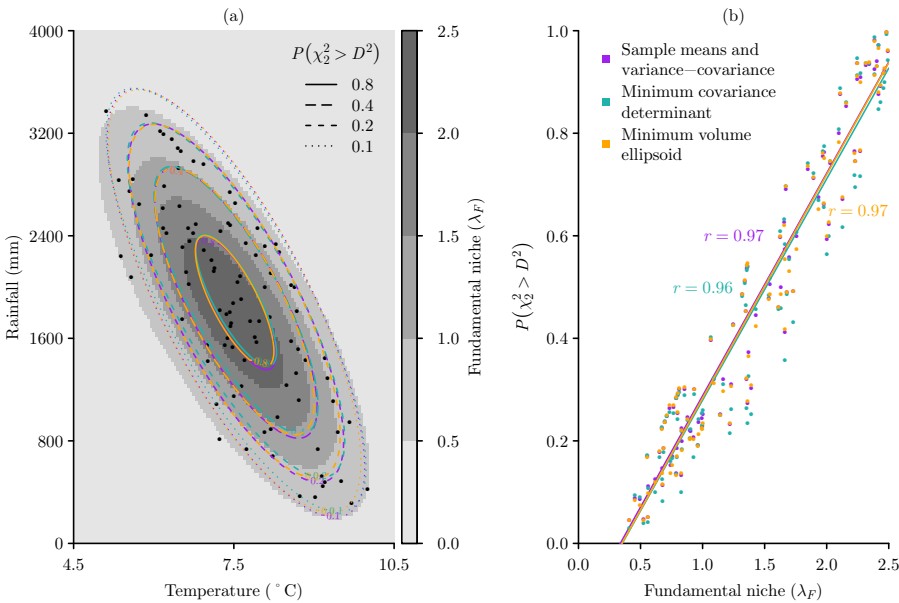

**Figure 2** **Modelling the fundamental niche $\lambda_F$ of a virtual species with Mahalanobis distances $D^2$ based on three multivariate location and scatter methods.** (A) Given an idealised sample of 100 occurrences across the virtual fundamental niche, the probabilities $P(\chi_2^2 > D^2)$ of the environmental space being within the niche are calculated for three multivariate location and scatter methods: the sample means and variance-covariance, the minimum covariance determinant, and the minimum volume ellipsoid. (B) With this idealised sample all three methods perform very well, producing very high correlations between the sample occurrences known $\lambda_F$ and estimated $P(\chi_2^2 > D^2)$.

which $P(\chi_2^2 > D^2)$ estimates for all three methods of multivariate location and scatter align with the elliptical shape of the fundamental niche (Fig. 2A). Under these idealised sampling conditions we can see that the actual $\lambda_F$ values and the estimated niche probabilities $P(\chi_2^2 > D^2)$ for the occurrence samples are very highly correlated, and that all three methods of determining the multivariate location and scatter of the niche perform equally well (Fig. 2B).

Of course this idealised virtual sampling is completely unrealistic, but does serve to demonstrate that it is possible to produce useful models with enough accurate data. What is of interest is understanding under what conditions of sample size, error, and bias the predictive ability of the various multivariate location and scatter methods begin to break down. Therefore, to explore this, a series of virtual experiments were conducted.

All experiments were done using R (*R Core Team, 2019*) with the MASS (*Venables & Ripley, 2002*), virtualNicheR (*Etherington & Omondiagbe, 2019*), fields (*Nychka et al., 2017*), raster (*Hijmans, 2020*), and extrafont (*Chang, 2014*) packages.

## Sample size

It is common for many species to have as few as seven unique occurrence locations within the Global Biodiversity Information Facility (*Meyer, Weigelt & Kreft, 2016*). However, as the MCD approach needs $m \geq n \times 5$ (*Hubert & Debruyne, 2010*), and given the experiments are two-dimensional, the smallest value of $m$ that could be analysed is 10. Therefore, to

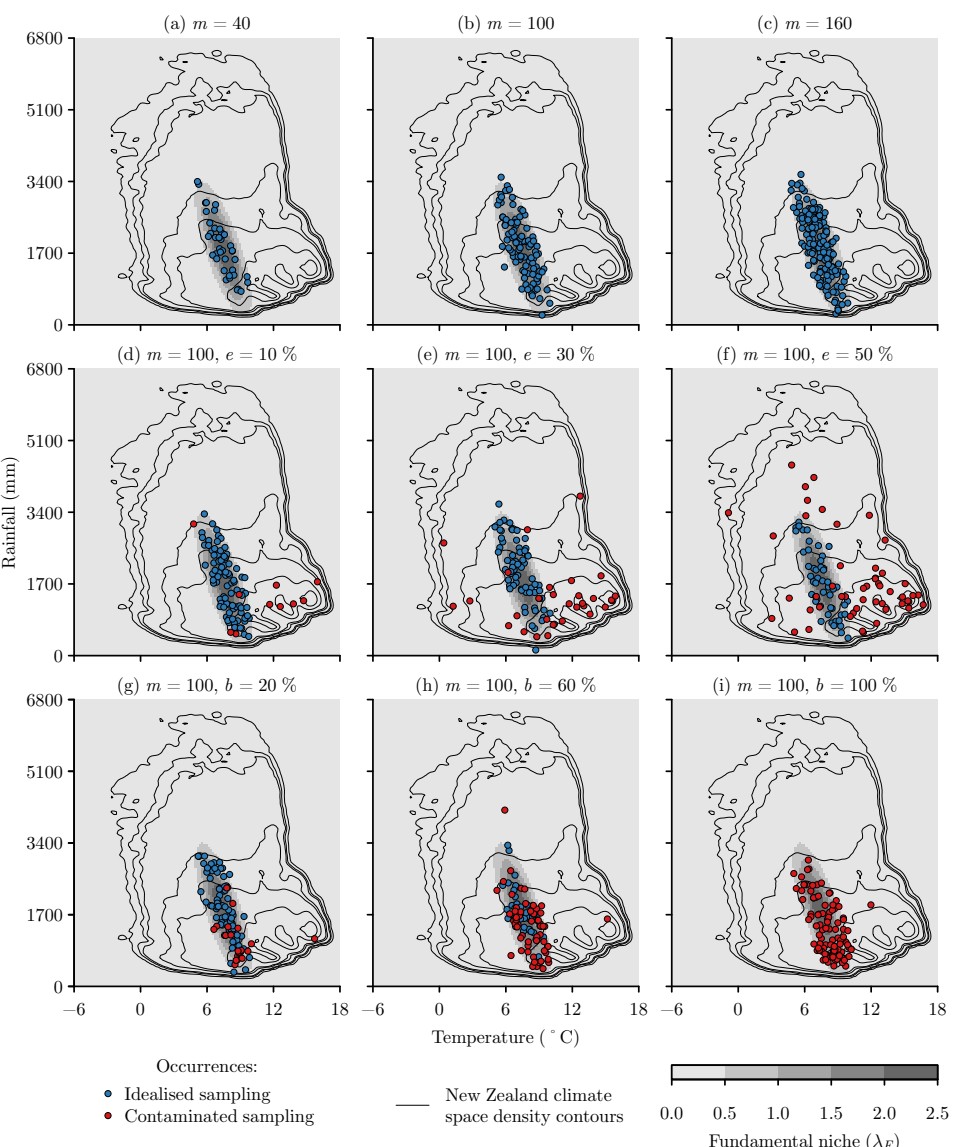

**Figure 3 Examples of virtual species occurrence samples formed by idealised and contaminated sampling at varying levels of sample size ($m$), error ($e$), and bias ($b$).** Idealised samples of (A) $m = 40$, (B) $m = 100$, and (C) $m = 160$ follow the elliptical shape of the fundamental niche with a greater intensity of occurrences towards the centre of the niche. Contaminated sampling was based upon the climate space of New Zealand, with increasing errors of (D) $e = 10\%$, (E) $e = 30\%$, and (F) $e = 50\%$ producing samples that increasingly represent the climate space rather than the niche, and increasing bias of (G) $b = 20\%$, (H) $b = 60\%$, and (I) $b = 100\%$ producing samples that increasingly represent the overlap between the climate space and the niche.

explore the effects of sample size, the idealised sampling approach was applied, but varying the sample size $m$ from 10 to 160 in increments of 15. As $m$ increases, the idealised samples increasingly represent the elliptical shape of the fundamental niche (Figs. 3A–3C)

Regarding the choice of sample subset $h$ used by the MCD and MVE methods, the standard choice is $h = \lfloor (m + n + 1)/2 \rfloor$ for both the MCD and MVE methods, because

this produces the most robust estimates (*Hubert & Debruyne, 2010*; *Van Aelst & Rousseeuw, 2009*). This standard choice uses just over half the sample, so for sample sizes from 10 to 160 this would mean a standard choice of $h$ from 6 to 81 as the sample size increases. Because $h$ varies as a function of sample size, for experimental consistency and to aid interpretation I specified the sample subset used by the MCD and MVE methods as a proportion $k$ of the sample size, such that $h = \lfloor k \times m \rfloor$. The standard choice of $h$ was represented by $k = 0.55$, and with $k = 0.75$ and $k = 0.95$ used to explore the effect of increasing $k$, and therefore $h$, on the performance of the MCD and MVE methods.

Each sample of size $m$ was replicated 500 times, and was applied to each multivariate location and scatter method, with the MCD and MVE methods applied at the three different $k$ values. The performance of each multivariate location and scatter method was measured as the correlation between the actual $\lambda_F$ and estimated $P(\chi_2^2 > D^2)$ values.

### Sample error

The virtual experiments to explore the effects of sample error followed the same process as the sample size experiments except that the sample size was fixed at $m = 100$, and each sample was contaminated with various levels of errors.

In generating errors I assumed that extreme errors can be identified using commonly used data checking processes (*Zizka et al., 2020*), so errors were limited to locations within New Zealand that comprise the core range of our virtual species. The climate space of New Zealand was described in terms of mean annual temperature and annual precipitation climatologies for the period 1979-2013 with a 30 arc second (around 1 km) grid resolution (*Karger et al., 2017*). An error was generated simply as a random location within New Zealand, and the amount of error within each sample was varied from 0% to 50% in increments of 5%. As error increases, the samples with error reflect the fundamental niche less and the New Zealand climate space more (Figs. 3D–3F).

### Sample bias

The sample bias experiments followed the same process as the sample error experiments, except that when selecting a random location within New Zealand the probability of the species being both present and detected (Eq. 5) was applied to limit bias to environments that are part of the fundamental niche. The amount of bias within each sample was varied from 0% to 100% in increments of 10%. As bias increases, the biased samples become concentrated at the more commonly occurring climate space of New Zealand that overlaps with the fundamental niche (Figs. 3G–3I).

## RESULTS

There were obvious consistencies and trends amongst all the experiments. First, the results for the MCD and MVE methods were very similar, making it hard to differentiate the performance of these two methods (Figs. 4–6). Second, in all cases, as $k$ increases the MCD and MVE methods become more similar to the sample means and variance-covariance method (Figs. 4–6).

Regarding the effects of sample size, the sample means and variance-covariance method performed better, but this difference only became notable when $m \lesssim 50$ and was less
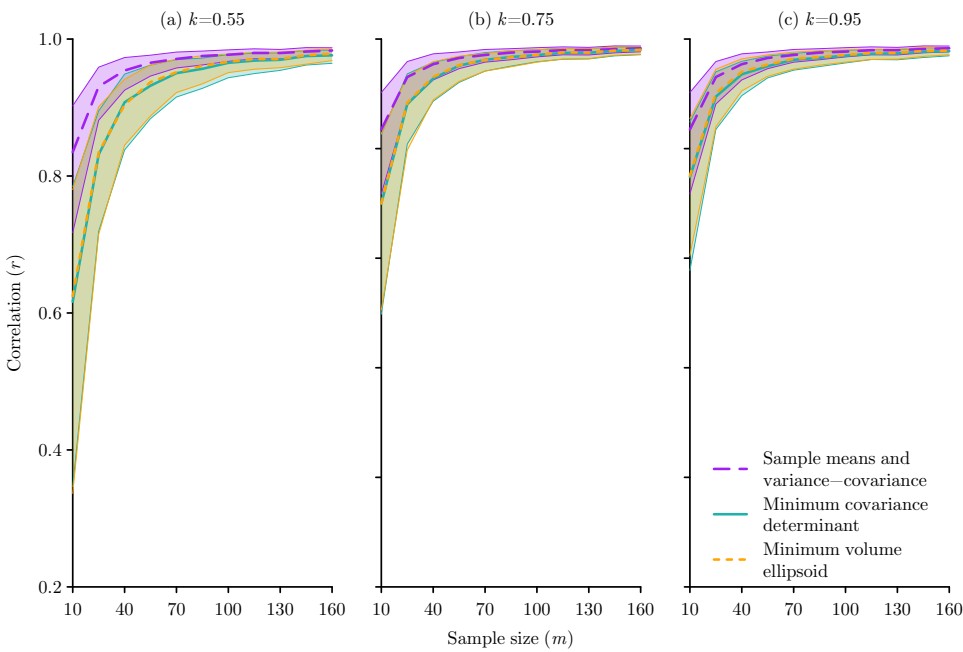

**Figure 4  The effect of species occurrence sample size on the performance of Mahalanobis distance niche models based on three different multivariate location and scatter methods: sample means and variance-covariance, minimum covariance determinant, and minimum volume ellipsoid.** The median and inter-quartile range of the correlation between the known niche value and the Mahalanobis distance probability for the occurrence sample from 500 replications are plotted for each method. The proportion $k$ of occurrences used for the minimum covariance determinant and the minimum volume ellipsoid were set at (A) $k = 0.55$, (B) $k = 0.75$, and (C) $k = 0.95$.

pronounced as $k$ increased (Fig. 4). The fact that $m = 100$ gives good results regardless of the method is important to recognise, because as we can be confident that any performance effects in the error and bias experiments that used $m = 100$ will be a function of the imposed error or bias rather than the sample size.

Considering errors, in all cases the MCD and MVE methods performed better than the sample means and variance-covariance method, though this difference was only evident at lower error levels for higher $k$ values (Fig. 5). The pattern of response to bias was more subtle, with the sample means and variance-covariance method performing better when bias was $\gtrsim 50\%$, but with this effect being very prominent when $k = 0.55$, less obvious when $k = 0.75$, and no longer present when $k = 0.95$ (Fig. 6).

## DISCUSSION
### Methodological differences
Errors and bias are inevitable in natural history data (*Graham et al., 2004*), and it is unlikely that an error-free and unbiased set of data can be produced. Also, as the fundamental niche becomes less similar to the sampling space, the potential effect of errors and bias should increase. This means that ecological niche modellers must consider how to minimise the effects of error and bias in their analyses. For those modellers using $D^2$ for niche modelling
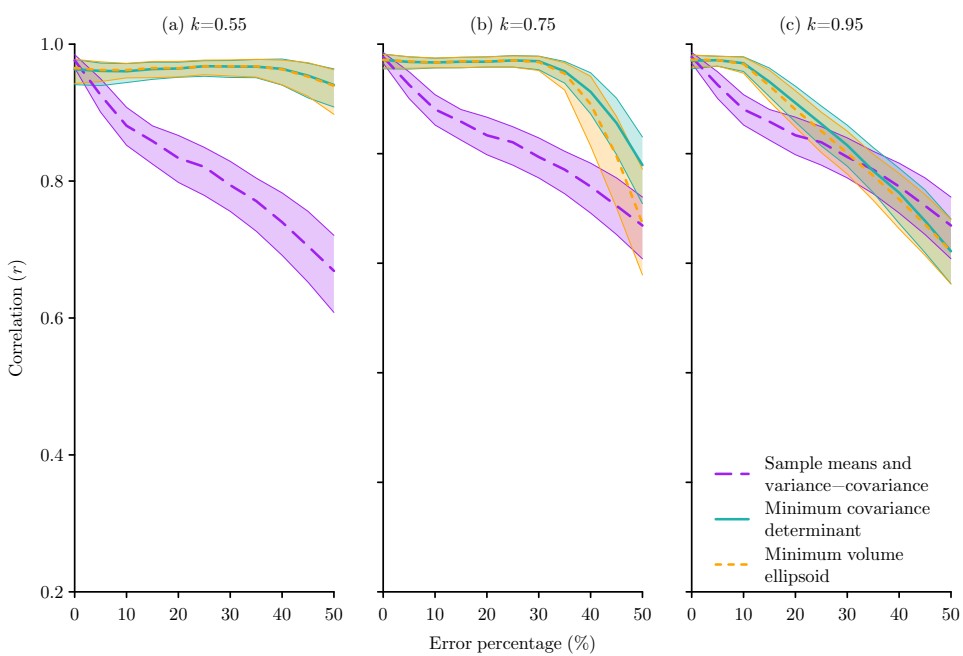

**Figure 5** **The effect of species occurrence sample errors on the performance of Mahalanobis distance niche models based on three different multivariate location and scatter methods: sample means and variance-covariance, minimum covariance determinant, and minimum volume ellipsoid.** The median and inter-quartile range of the correlation between the known niche value and the Mahalanobis distance probability for the occurrence sample from 500 replications are plotted for each method. The proportion $k$ of occurrences used for the minimum covariance determinant and the minimum volume ellipsoid were set at (A) $k = 0.55$, (B) $k = 0.75$, and (C) $k = 0.95$.

or outlier detection, the results from these virtual ecology experiments demonstrate that MCD and MVE multivariate location and scatter methods provide an opportunity to avoid the influence of errors, but this must be balanced against the influence of bias. Therefore, modellers more concerned about the prevalence of errors should choose lower values for $k$, while modellers more concerned about the prevalence of bias should choose higher values for $k$. This finding supports the advice of *Qiao, Soberón & Peterson (2015)*, who advise that there is no optimal modelling approach, and that choices need to be made based on the individual data and question.

As $h$ is a free parameter that can vary $\lfloor(m+n+1)/2\rfloor \leq h \leq m$ (*Hubert & Debruyne, 2010*), or in proportional terms $0.5 \lesssim k \leq 1$, it is critical that users of both MCD and MVE clearly report the value of $h$ or $k$ used. Specifying $h$ or $k$ is important for interpreting results, because when $h \to m$ or $k \to 1$ the MCD method becomes equivalent to the conventional sample means and variance-covariance method, and the MVE method produces ever larger ellipses that will eventually encapsulate all the data (*Rousseeuw, 1985*). Therefore, studies that do not specify the $h$ or $k$ parameter (*Norris, Jackson & Betancourt, 2006*; *Liu, White & Newell, 2018*; *Qiao et al., 2019*) are not accurately reporting their methods. Also, given that the default value in software such as MASS (*Venables & Ripley, 2002*) is very close to $k = 0.55$, then based on the results here, if authors are not reporting a choice of $h$ or $k$

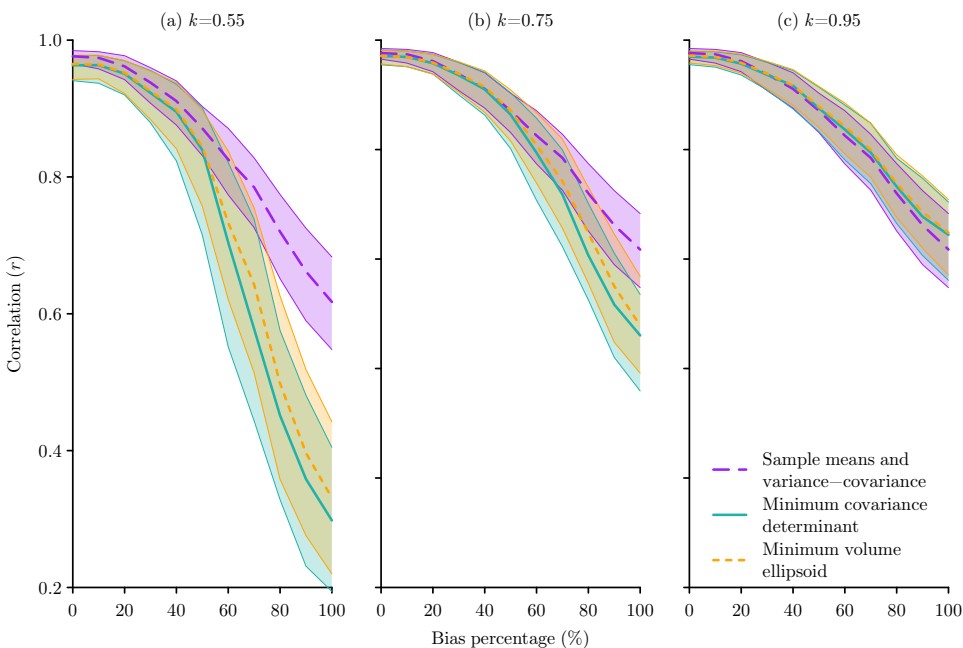

**Figure 6** **The effect of species occurrence sample bias on the performance of Mahalanobis distance niche models based on three different multivariate location and scatter methods: sample means and variance-covariance, minimum covariance determinant, and minimum volume ellipsoid.** The median and inter-quartile range of the correlation between the known niche value and the Mahalanobis distance probability for the occurrence sample from 500 replications are plotted for each method. The proportion $k$ of occurrences used for the minimum covariance determinant and the minimum volume ellipsoid were set at (A) $k = 0.55$, (B) $k = 0.75$, and (C) $k = 0.95$.

because they are relying on the default value, then they are potentially applying methods that do not perform well with the small sample sizes (Fig. 4A) and high levels of bias (Fig. 6A) that are prevalent in natural history data (*Meyer, Weigelt & Kreft, 2016*). This supports the statement of *Peterson et al. (2011)* p. 113 that "it is generally poor practice to use default settings provided by software without justification, testing, and exploration of these values for a particular application", and hopefully the results presented here can provide some guidance for choosing the $h$ value.

Those studies that have reported their choice of parameter have used values of $k = 0.95$ (*Soberón, Peterson & Osorio-Olvera, 2018*; *Altamiranda-Saavedra et al., 2020*; *Castaño-Quintero et al., 2020*), $k = 0.975$ (*Osorio-Olvera et al., 2020*; *Castaño-Quintero et al., 2020*), and $k = 0.99$ (*Yañez-Arenas et al., 2018*). These seem to be sensible choices based on the virtual experiments conducted here, as in comparison to the sample means and variance-covariance method, when $k = 0.95$ the MCD and MVE methods will not be too negatively affected by small sample sizes (Fig. 4C), are an improvement when errors $\lesssim 20\%$ (Fig. 5C), and will not perform worse at any level of bias (Fig. 6C).

In terms of choosing between MCD and MVE, while both were introduced simultaneously (*Rousseeuw, 1985*), MVE was initially more readily used due to its computational simplicity, but with the development of better algorithms MCD has

been suggested as the preferred option due to its statistical efficiency (*Rousseeuw & Van Driessen, 1999*). However, the results of the experiments conducted here might indicate a slight preference for the MVE method as the MCD only performed slightly better in the error tests when the error percentage was above 30% (Fig. 5B), which is probably unrealistically large, while the MVE performed slightly better for all the bias tests (Fig. 6). Ultimately I do not think there is much difference between the MCD and MVE methods, so the choice of either for ecological niche modelling is equally justifiable.

## Reducing errors and bias in natural history observation data

The advantage of virtual experiments is that we can know the exact conditions of the data, but in reality the actual levels of error and bias remain unknown and can only be estimated based on experience with the data. This makes it hard to choose which multivariate location and scatter method is optimal in any given situation. However, the performance of all the multivariate location and scatter methods will improve with reduced errors and bias, so all ecological niche modellers should give this serious consideration.

Automated approaches can be tailored to rapidly detect likely errors within natural history observation data by checking for internal consistency of the meta-data and by comparison with complementary datasets (*Zizka et al., 2020*). In contrast, bias is harder to detect and correct (*Graham et al., 2004*), but is almost guaranteed to exist as this study has shown that even random sampling in geographical space leads to a biased sample of a niche in climatic space. This finding is supported by other research that also demonstrated that sampling bias is likely to become even worse as geographic sampling is further constrained to more accessible areas such as around roads (*Albert et al., 2010*) that is a common feature of natural history observation data (*Reddy & Dávalos, 2003*). This issue of bias is of particular importance for presence-only methods such as $D^2$ that are particularly attractive when working with natural history observation data that have no absences and that are sufficiently unstructured to reliably define the background environmental data (*Etherington et al., 2009*). However, while presence-only techniques have minimal data requirements, it becomes harder to detect and manage bias because the absence and background data can provide useful contextual information. When absence data are available, then presence-absence methods such as logistic regression may be expected to suffer less from bias, because biases in presence data can be balanced out by similar biases in absence data (*Zadrozny, 2004*). Similarly, with presence-background methods, the background data can be created to have similar bias to the presence data to minimise the effects of bias (*Phillips et al., 2009*). Effective bias reduction options for presence-only methods include spatial filtering to reduce the intensity of the bias, either in geographic (*Boria et al., 2014*) or more optimally in environmental space (*Varela et al., 2014*; *Castellanos et al., 2019*). Spatial filtering has been shown to be more effective than background manipulation for presence-background methods (*Kramer-Schadt et al., 2013*), and so it should be an effective bias reduction technique for presence-only methods such as $D^2$, assuming the filtered sample sizes do not become problematically small (Fig. 4).

In summary, there are methods to reduce error and bias, but what level of errors and bias remain will be unknown. Therefore, given bias is harder to detect than errors, and that

there are reduced options to control bias for presence-only models, I would suggest that ecological niche modellers err towards multivariate location and scatter methods that are less sensitive to bias.

## The assumption of normality

Regardless of the choice of multivariate location and scatter method used to calculate $D^2$, it is important to consider if the fundamental niche can be reasonably approximated by the elliptical shapes resulting from the underlying multi-dimensional normal distribution. Field studies of abundance or occurrence along environmental gradients have shown some normally distributed species responses, but most responses, while unimodal, are skewed, and some even show bimodal responses (*Whittaker, 1952*; *Whittaker, 1956*; *Whittaker, 1960*; *Terborgh, 1971*; *Austin, 1987*). However, we need to recognise that it is ultimately impossible to truly measure the fundamental niche in the real world, as biotic interactions mean that only the realised niche can be measured, and realised niches may well take on very complex shapes that are quite different to the fundamental niche (*Austin & Smith, 1989*; *Blonder, 2016*; *Soberón & Peterson, 2020*). In fact, the real world situation is even more limited because the environmental space that can be sampled is actually a complex interaction of the environments that currently exist, biotic interactions, and dispersal limitations (*Soberón & Peterson, 2005*), and even the view of this limited environmental space is warped by the sampling bias inherent in natural history observation data (*Meyer, Weigelt & Kreft, 2016*). Given these complexities of sampling from the real world, the fundamental niche can only really be measured through experimental manipulations. However, there is very little of this experimental evidence (*Soberón & Peterson, 2020*), so the expected shapes of fundamental niches remain unclear.

Ultimately, the inability to collect normally distributed data from the real world does not preclude the use of $D^2$ as a fundamental niche model. Means and variances can be calculated from any distribution of data, and the fact that $D^2$ models compare favourably with other modelling approaches (*Dettmers, Buehler & Bartlett, 2002*; *Johnson & Gillingham, 2005*; *Tsoar et al., 2007*) suggests the fundamental niche can be approximated as elliptical in at least some settings. Even when it is not desirable to assume a fundamental niche is normally distributed, $D^2$ can still be a used to eliminate outliers. This could support other fundamental niche modelling methods, such as convex hulls (*Pironon et al., 2019*), that are not limited to the assumption of normality but are sensitive to outliers (*Blonder, 2018*).

## CONCLUSIONS

When using $D^2$ for ecological niche modelling and outlier detection, the performance of multivariate location and scatter methods varies based on sample size, error, and bias. Comparison of the sample means and variance-covariance, MCD, and MVE multivariate location and scatter methods provides the clear conclusion that none of the methods, or individual parameterisations of any methods, can be considered universally the best. Rather, any ecological niche modeller using these techniques needs to think carefully about their data and objective to choose the method and parameterisation that are most appropriate to their individual circumstances. For those modellers who wish to explore

the potential of the MCD and MVE methods, given these methods have been used widely in statistical analyses for some time, these methods should be widely available in statistical software. However, modellers using the MCD and MVE methods should carefully consider and clearly state the $h$ or $k$ parameter used in their analyses.

### Funding

This research was supported by the Winning Against Wildings Research Programme funded by the New Zealand Ministry of Business, Innovation and Employment. The funders had no role in study design, data collection and analysis, decision to publish, or preparation of the manuscript.

### Grant Disclosures

The following grant information was disclosed by the author:
New Zealand Ministry of Business, Innovation and Employment.

### Competing Interests

Thomas R. Etherington is employed by Manaaki Whenua-Landcare Research, and declares that he has no competing interests.

### Author Contributions

- Thomas R. Etherington conceived and designed the experiments, performed the experiments, analyzed the data, prepared figures and/or tables, authored or reviewed drafts of the paper, and approved the final draft.

### Data Availability

Data and code used to conduct and plot the results of the virtual experiments are available as Supplemental Material.

### Supplemental Information

Supplemental information for this article can be found online at http://dx.doi.org/10.7717/peerj.11436#supplemental-information.

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
