# Peer review of "Mahalanobis distances for ecological niche modelling and outlier detection: implications of sample size, error, and bias for selecting and parameterising a multivariate location and scatter method"

_PeerJ, doi:10.7717/peerj.11436_

## Round 0.1 · original submission · Major Revisions

· Academic Editor

Major Revisions

All reviewers appreciated your work, and I fully support their assessment. They also make constructive comments that I believe can improve the paper. In particular, you should perhaps reconsider your use of "detection" (l. 115, eq 5 and later), as many would take it as the probability of observing the species given it is present (as in occupancy models), whereas (if I understood it correctly), it represents the probability that the species is in fact present (and therefore observed in your model).
It would also be relevant to comment further on sampling design, as emphasized by one reviewer. Some years ago, we commented on various designs to estimate niche models (Albert et al. 2010 Ecography), and I think your study sheds some light on this issue. The reviewer provides relevant references so you do not need to cite our paper.

Reviewer 1 ·

Basic reporting

As a whole, I found this paper well written, concise and interesting. The simulations help the reader to assess how the method is affected by less-than-ideal conditions. I think however that the discussion section could be improved. In particular, I would suggest to the author to improve the discussion of the results in the light of previous work carried out on this issue.

On figures 4, 5, 6. There is a large amount of white space on these figures, which makes the plots difficult to read. The author should change the y-limits of the plot to focus on the curve. I understand that the author wanted to include 0/1 limits for the correlation, but including the value 0 here is not necessary.

Moreover, on figure 4, the results for MCD are difficult to see (the green curve is behind the orange curve and can only be seen by zooming on the curve). Is it possible to improve readability of this plot (e.g. using dashed lines)?

Experimental design

No comment.

Validity of the findings

No comment

Additional comments

This paper uses a simulation approach to compare three methods to estimate the mean vector and covariance matrix of the niche when Mahalanobis distances are used for ecological niche modelling. The author assesses the effect of the sample size, error and bias on the results of the three methods, also varying the proportion of the sample used by the MCD/MVE methods to estimate the parameters of the niche. He concludes that the MCD and MVE methods can be useful to correct random errors when the sample size is large enough and when the bias is not too large. He also stresses the importance to report the parameters h and k when the methods MCD and MVE are used, and give some advice on how to set them.

As a whole, I found this paper well-written, concise and interesting. The simulations help the reader to assess how the method is affected by less-than-ideal conditions. I think however that the discussion section could be improved. In particular, I would suggest to the author to improve the discussion of the results in the light of previous work carried out on this issue. More precisely:

1. A large part of the discussion presently focuses on the normality assumption, though this aspect *per se* is not really studied in the paper (although indirectly through the study of bias): the author did not test how the shape of the niche affects the method used, so that -- even if this discussion is interesting -- it should be shortened on this specific question, and be more related to the results of the simulations (see next points).

2. I really liked how the author tested the effect of increasing bias in sampling on the Mahalanobis distances. In my opinion, this is a very clear illustration of a point that will not be immediately clear to all readers: the author simulated a bias *by sampling randomly locations in the geographical space*: this resulted in an increased sampling intensity in environmental conditions more common in the study area. Of course, it is clear that when a sample is used to model a statistical distribution, the data collection must be considered ignorable in the model. However, in the context of niche modelling, not all users will understand that a random sample in the geographical space will often not be ignorable, resulting in a biased estimation of the niche parameters. Indeed, for other niche modelling methods used to estimate species distribution maps, it is generally recommended to sample the whole area of interest randomly (e.g., from Royle et al. 2012, Meth. Ecol. Evol., 3, 545--554: "*We emphasize the critical assumption required for statistical inference about species occurrence probability from presence-only data, which is random sampling of space as a basis for accumulating presence-only observations*". See also Kramer-Schadt et al 2013, Diversity and Distribution, 19, 1366-1379), since other methods basically try to separate presence and absence of the species on the area to model the niche shape. It is in my opinion important to stress that the apparent benefit of relying on presence-only data is only gained at the expense of collecting an unbiased sample from the ecological space (which will be rare with the type of natural history data considered in the introduction of the paper). Though attractive in theory, Mahalanobis distances as such may be of limited use when the sampling does not allow an unbiased estimation of the niche parameters (which will generally be the case), and MCD/MVE will often be a band-aid on a wooden leg when the bias is important, as clearly demonstrated by the simulations.

3. Note that, on this concept of bias, it may be interesting to relate these results to the approaches of Knick and Rottenberry (1998, JABES, 3, 311--322, who also studies the effect of biased sample on the D2), and the suggested solutions by Rotenberry et al. (2006, Ecology, 87, 1458-1464) and Calenge et al. (2008, Ecology, 89, 555-566). Note that their approach could also -- in theory -- be used with MCD/MVE estimated mean vectors and covariance matrices.

4. An important issue to be discussed in the discussion is therefore how unbiased data can be collected to calculate D2 (and how this sampling differs from other niche modelling method). In his simulations, the author supposed a multinormal sampling of the environmental space with the same parameters as those of the niche, noting that this is unrealistic. In other words, under this model, unbiased data for the estimation of $\mu$ and $\Sigma$ can only be collected when $\mu$ and $\Sigma$ are already been known. What does the author recommends in real conditions? Should the D2, even with MCD/MVE parameters estimation, be used at all with such data?

Some detailed comments follow:

line 106: I suggest to be more precise and replace "the finite rate of increase" with "the finite rate of increase lambda_f(x) of a population of the focus species in places with environmental characteristics x, where x is a vector containing the temperature and rainfall conditions".

line 115: please replace "the probability of detection" with "the probability that a population of the species is both present and detected in places with environmental conditions x" (to avoid confusion with what is usually termed "probability of detection", i.e. the conditional probability to detect the population given that it is present).

Equation (5): improve notation: replace P(d) with P_d(x) and \lambda_F, with \lambda_{F(x)}.

lines 113: I may miss something here, but how the assumption of a random sample from a normal distribution with mean \mu and covariance matrix \Sigma in the environmental space, followed by a subsample of this sample with a probability calculated with eq. 5 leads to unbiased presence data sampled from the niche? I can see on fig. 2b that the simulated value of \lambda_f is well correlated with $P(\chi^2_2 > D^2)$, but is there any theoretical argument for this result ? (or reference : Etherington and Omongdiagbe, 2019, cited in the paper, does not describe it either).

Reviewer 2 ·

Basic reporting

This paper explores the utility of Mahalanobis Distance (D^2) within ecological niche modelling and compares three methods for defining multivariate location (\mu) and scatter (\Sigma):
(1) Sample mean and covariance
(2) Minimum covariance determinant (MCD)
(3) Minimum volume ellipsoid (MVE).
Exploring these three approaches is well motivated by the prevalence of errors and sampling bias found in natural history data and this motivation is well translated into the difference scenarios examined. However, the methodology for how samples (m) are simulated would benefit from further clarification and explanation. Several terms used do not match with how they are conventionally used in the literature. For example, on line 117 the author outlines how the probability of detection is calculated. To me, detection probability implies given a species is present in a location, there is uncertainty in whether it is observed. However, since this is calculated based on population growth rate (\lambda_F) it seems to be more akin to a probability of species presence/occupancy? Ultimately, further clarification including an explicitly written out model at all levels is needed. More details may be found in included annotated PDF.
The literature well supports the use of MD as a measure useful in both outlier detection and niche modelling. A notable extension of this work to identifying extrapolation within these same niche models (i.e. species distribution models) is missing (e.g. Here be dragons: a tool for quantifying novelty due to covariate range and correlation change when projecting species distribution models. 2014. Mohsen B. Mesgaran, Roger D. Cousens, Bruce L. Webber) and would be a useful inclusion.
The article structure is clear. Figures are relevant, informative and clear.
I commend the author on their very well organized and clearly commented supplementary materials. Data acquisition, simulations, and figures are all straight forward to reproducible.

Experimental design

These simulations are clearly motivated, well reported, and easily repeated. Figure 3 is very handy for visualizing two of the three outlined scenarios. Adding an additional row providing a snapshot of different sample sizes (including m=100) might make it more representative of all three scenarios.

Validity of the findings

The findings are reported in an orderly fashion, link with the aims and objectives of the paper, and are connected back to previous literature’s findings where applicable. The reporting is well supported by Figures. Recommendations are clearly provided for future applications.
I question the requirement of normality underlying the use of MD for niche modelling. While it is true that multivariate normal data provides the convenience of D^2 following a chi-squared distribution, without this assumption you still have a measure of distance from a centroid location within some multivariate covariate space. It is still possible select some boundary(s) to guide how likely a location is within a more complex niche. I suspect that adapting the used in these simulations might require this normality assumption code (e.g. MASS::cov.rob() function), but in a more general context MD can be use without any underlying distribution assumption.

Additional comments

This is a well-written article and I enjoyed reading it for this review. Please see attached annotated PDF for additional comments. I hope my comments are clear and useful.

Annotated reviews are not available for download in order to protect the identity of reviewers who chose to remain anonymous.

·

Basic reporting

Before I provide specific comments on the manuscript, I want to highlight some general aspects that I consider should be included in a future version of the manuscript:
a) The manuscript needs a general check for long sentences. I highlighted specific paragraphs in the comments below, where adding a comma or re-phrasing the sentence could help convey the message better.
b) I suggest including a definition of sample error and sampling bias and give examples of each of them. In the current version of the manuscript, it is not clear what are the sources of each and why are there relevant. This can be done either in the Introduction section or the corresponding subsections in Materials & Methods where the author explained how these phenomena were included in the simulations.
c) I suggest adding an explanation of the meaning of the parameters of interest (similar to what the author included in the lines 108-109), mu and Sigma, regarding their relationship to the fundamental niche of a species in the Introduction, where the Mahalanobis distance is defined.
d) The author only explored one set of values for the parameters of interest (mu and Sigma) which I think is enough to illustrate very well the central ideas of the study; however, by seeing Figure 1, it seems that the position of the center of the ellipses regarding the climate space defined by the area of study could influence the results from different methods. So, what would it happen with your results if the center of the ellipse is either closer or farther, to the highest concentration of points in climate space? Would you expect better/worse results under these scenarios? Maybe this can be added to the Discussion.

Experimental design

Overall, I think that the simulations presented in the manuscript were well designed and help illustrate important aspects that the users need to consider when using these methods.

Validity of the findings

In this manuscript, the author compared the performance of three methods to estimate the vector of means and the covariance matrix that defines a Mahalanobis distance in an ecological niche modeling context: the sample mean and variance-covariance, the minimum covariance determinant, and the minimum volume ellipsoid. With the purpose of providing recommendations regarding the selection and parametrization of these methods, the author explored scenarios that simulate different levels of error and bias (known to occur in real samples) together with different values for the sample size of species occurrences. These methods have been used in many recent papers without providing a justification of why it is reasonable to choose one of them among the others, therefore, I consider that the findings of the author will have significant impacts in the field of ecological niche modeling. Finally, the discussion on the assumption of normality will be central for future applications of the explored methods in ecological niche modeling.

Additional comments

Specific comments:
Lines 37-40: the first part of the sentence contains information that was already mentioned in the previous lines of the abstract, perhaps make the whole sentence shorter? For example: "It is critical that users of the minimum covariance determinant and minimum volume ellipsoid methods clearly report their parametrizations so the results can be interpreted correctly."
Line 48: add “(occurrences)” after “species observations”
Lines 83 and 91: add a comma after “(Figure 1)”
Line 90: replace “However” by “On the other hand” (or equivalent phrase)
Lines 105-106: provide a reference
Line 111: it is not clear what the fundamental niche rate of increase means, please explain in more detail
Line 114: “occurrences, initially I assumed” instead of “occurrences I assumed”.
Line 115-118: what is the relationship between the probability of detection and sampling bias? Explain either here or in the Sampling bias subsection.
Lines 118-120: difficult to read, rephrase it
Line 138-145: difficult to read, rephrase it
Lines 170-174: add a reference to specific figures after each sentence, also, the whole paragraph is difficult to read, rephrase it
Line 177-179: difficult to read, rephrase it
Lines 194-196: is it possible that users have a sense of the magnitude of sampling bias and sampling error present in their samples? In which situations should the user be more concerned about the prevalence of sampling error/sampling bias?
Line 260-264: difficult to read, rephrase it

Figures and Tables:
I suggest adding labels or legends to the different panels of Figures 3, 4, 5, and 6 that allow the reader to identify the different scenarios without having to look for the captions of the figure.
In Figure 1, the outlier is difficult to spot. I noticed its existence until it was mentioned in the text. I suggest modifying the figure, maybe by increasing the length of the axes so the outlier does not fall close to the margins of the figure.

---

## Round 0.2 · Minor Revisions

· Academic Editor

Minor Revisions

All reviewers very much appreciated the efforts you put in revising your manuscript and I fully agree. One reviewer suggested some minor changes that you should consider in the final version of the paper. This is a good paper representing a useful contribution to the field.

Reviewer 1 ·

Basic reporting

The author has taken all my comments into account. He has made the M&M clearer and deeply reworked his discussion of the results. In my opinion, he took into account all my suggestions satisfactorily. I just have three final minor suggestions (see comments to authors).

Experimental design

no comment

Validity of the findings

no comment

Additional comments

My three suggestions are the following:

* I really appreciated the amount of work that the author has done by writing the new subsection "Reducing errors and bias" of the discussion, and in particular the comparison of presence-only vs. presence/absence data. However, since this discussion focuses on natural history observation data, and does not consider data collected specifically for niche modelling, the author should change the title of the subsection to "Reducing errors and bias in natural history observation data".

* line 267 (after "However, while presence-only techniques have minimal data requirements, it becomes harder to detect and manage bias because the absence and background data can provide useful contextual information"), I think that it would be important to stress *explicitly* that your study illustrates clearly how a random sample in the geographical space will in most cases result in a biased inference of the niche with the MD approach. The paper of Albert et al. (2010) stressed by the editor is indeed an interesting reference for this point: Albert et al. showed the importance to sample in the appropriate sampling space: a random sampling in geographical space will not lead to a representative sample in the climatic space, and your study illustrates clearly that this will be of the utmost importance for MDs. It would be interesting to stress this point, even if this section focuses on natural history data (just two sentences to explain explicitly this point would be enough and add a lot of value to this discussion imho).

* A very minor point regarding the notation: I thank the author for taking into account my comment regarding the notation \lambda_f and P_d to \lambda_F(x) and P_d(xi) respectively. The author may want to put the "(x)" in normal font (i.e., not subscript -- only F and d would be in subscript) to make it clear that it is lambda (resp. P) that depends on x and not F (resp d).

Reviewer 2 ·

Basic reporting

This paper looks great. Revisions have been incorporated and addressed nicely.

Experimental design

No comment.

Validity of the findings

No comment.

Additional comments

It is clear that you considered and addressed each comment and provided clear responses and updates to the manuscript. Wonderful work!

·

Basic reporting

As a whole, I found the second version of this manuscript clear, well-written, and concise. The concepts of error and bias in occurrence samples are now clear and the author provides simulations that help the reader to assess how each method is affected by different levels of error and bias. The new version of the Discussion section provides insightful comments regarding the use of the three analyzed methods in ecological niche modeling. I am sure the readers will find this work interesting and useful. Finally, all the figures were improved and look neat.

Experimental design

no comment

Validity of the findings

no comment

Additional comments

All the comments I made to the original manuscript well satisfactorily addressed by the author and I have no further suggestions.

---

## Round 0.3 · accepted · Accept

· Academic Editor

Accept

Thanks for carefully revising the paper following the comments by the last reviewer. This is a very nice paper that I hope will influence researchers analysing species distributions and ecological niche.